# Effects of an Organic-Inorganic Hybrid Containing Allyl Benzoxazine and POSS on Thermal Properties and Flame Retardancy of Epoxy Resin

**DOI:** 10.3390/polym11050770

**Published:** 2019-05-01

**Authors:** Benben Liu, Huiling Wang, Xiaoyan Guo, Rongjie Yang, Xiangmei Li

**Affiliations:** School of Materials Science, Beijing Institute of Technology, 5 Zhongguancun South Street, Haidian District, Beijing 100081, China; liuben0309@163.com (B.L.); 3120181135@bit.edu.cn (H.W.); gxy@bit.edu.cn (X.G.); yrj@bit.edu.cn (R.Y.)

**Keywords:** benzoxazine, POSS, organic-inorganic hybrids, epoxy resin, thermal properties, flame retardancy

## Abstract

A novel organic-inorganic hybrid containing allyl benzoxazine and polyhedral oligomeric silsesquioxane (POSS) was synthesized by the thiol-ene (click) reaction. The benzoxazine (BOZ)-containing POSS (SPOSS-BOZ) copolymerized with benzoxazine/epoxy resin was used to prepare composites of SPOSS-PBZ-E nanocomposites(NPs). The polymerization behavior was monitored by FTIR and non-isothermal differential scanning calorimetry (DSC), which showed that the composites had completely cured with multiple polymerization mechanisms according to the oxazine ring-opening and epoxy resin (EP) polymerization. The thermal properties of the organic–inorganic polybenzoxazine (PBZ) nanocomposites were analyzed by DSC and thermogravimetric analysis (TGA). Furthermore, the X-ray diffraction analysis and the scanning electron microscopy (SEM) micrographs of the SPOSS-PBZ-E nanocomposites indicated that SPOSS was chemically incorporated into the hybrid nanocomposites in the size range of 80–200 nm. The flame retardancy of the benzoxazine epoxy resin composites was investigated by limiting oxygen index (LOI), UL 94 vertical burn test, and cone calorimeter tests. When the amount of SPOSS reached 10% or more, the vertical burning rating of the curing system arrived at V-1, and when the SPOSS-BOZ content reached 20 wt %, the thermal stability and flame retardancy of the material were both improved. Moreover, in the cone calorimeter testing, the addition of SPOSS-BOZ hindered the decomposition of the composites and led to a reduction in the peak heat release rate (pHRR), the average heat release rate (aHRR), and the total heat release (THR) values by about 20%, 25%, and 25%, respectively. The morphologies of the chars were also studied by SEM and energy dispersive X-ray spectroscopy (EDX), and the flame-retardant mechanism of POSS was mainly a condensed-phase flame retardant. The ceramic layer was formed by the enrichment of silicon on the char surface. When there are enough POSS nanoparticles, it can effectively protect the combustion of internal polymers.

## 1. Introduction

Benzoxazine has a stable aromatic ring structure and can be cross-linked under certain conditions to form a structure such as a phenolic resin. [1]. During curing, there is no need for strong acid catalysts. The monomers can be prepared from low-cost raw materials such as phenols, primary amines, and formaldehyde [2,3,4,5]. Compared with traditional phenolic resins, benzoxazines have many excellent properties as a class of high-performance thermal setting resins [6]. However, with the higher requirements imposed by the development of electronic technology on printed circuit board (PCB) material, some disadvantages of benzoxazines such as high ring-opening temperature, brittleness, insufficient heat resistance, and poor compatibility with composites also need to be further improved [7]. At the same time, epoxy resin is the most widely used PCB material. However, epoxy resin also decays the insulating property of the PCB. In light of the above disadvantages of benzoxazines and epoxy resin, an epoxy resin needs to be incorporated into the benzoxazines to obtain both good mechanical and thermal properties in the copolymers [8,9,10]. It has been shown that the modification of epoxy with benzoxazine not only improves flame retardancy, but also improves the mechanical properties [11]. The phenolic hydroxyl structure formed after the ring opening of the oxazine ring can react with the epoxy ring group to form benzoxazine/epoxy (BOZ/EP) cross-linking copolymerization, which is superior to the conventional single-system polymer. According to recent research, the introduction of a third component based on EP and BOZ composites, or the modification of either EP or BOZ, has received more attention [12].

Polyhedral oligomeric silsesquioxane (POSS) is a novel organic–inorganic hybrid filler with a cage structure, which is mainly composed of three-dimensional cage-like structures connected by Si–O–Si bonds and is very stable. The incorporation of POSS into benzoxazine monomers can improve the thermal, mechanical [13,14,15,16], and other unique properties [17,18] of polybenzoxazine (PBZ) resins. It consists of an inorganic siloxane group as the inner core and the organic groups as an outer layer; both organic components and inorganic POSS can display enhanced performance capability compared with that of their non-hybrid polymeric materials [19,20]. Moreover, they have well-defined structures, high-temperature stability, monodisperse molecular weights, and greater design flexibility relative to conventional fillers (e.g., clay, graphene, carbon nanotubes, and boron nitride). Generally, chemical copolymerization [21] and physical blending can incorporate the POSS into polymer materials. However, the linkage of covalent bonds [22] can avoid macro-phase separation between the polymer matrix and the POSS nanocomposites (NPs). Three approaches have been reported for the introduction of POSS into BOZ monomers, which contain mono-benzoxazine functionalized POSS and multi-benzoxazine functionalized POSS [23]. Lee Y J et al. [24] synthesized vinyl-terminated benzoxazine together with amine-containing POSS, formaldehyde, and phenol in THF solution at 90 °C to obtain POSS-BOZ. However, these synthetic steps are complicated and are performed under very harsh conditions, especially in the case of platinum as the catalyst, which requires a water-free environment. Therefore, considering the wide-ranging applications of organic–inorganic hybrids, it is necessary to develop a rapid, highly efficient method for the target-oriented synthesis of hybrid materials.

It has been increasingly recognized that “good” reactions for making POSS-related materials including the criteria of efficiency, versatility, and selectivity; therefore, “click” reactions are ideal tools [25]. Li et al. [26] reviewed the preparation of different functional group POSS nanocomposites using “click” reactions, where this concept has generated much interest in creatively preparing materials of choice. Additionally, the polymer chain composition and POSS surface chemistry can be tuned in a modular and efficient way by thiol–ene “click” chemistry [27,28]. Furthermore, sulfur-based POSS can be introduced into olefin material [29] or epoxy resin [30] by thiol-ene and thiol-epoxy click reactions, respectively, and they have excellent dispersibility. Wu and Kuo [31] synthesized POSS-BOZ nanocomposites with octa-azido functioned POSS(OVBN_3_-POSS) and 3,4-dihydro-3-(prop-2-ynyl)-2H-benzoxazine (P-pa) by a click reaction. However, the preparation of octa-azido functionalized POSS is too complicated. In previous studies, no researchers used thiol-based POSS and allyl benzoxazine to obtain POSS-BOZ nanocomposites nor did they study their effects on the thermal stability and flame retardancy of BOZ/E composites.

Therefore, in this paper, an organic–inorganic hybrid benzoxazine was prepared by the thiol-ene (click) reaction between benzoxazine and sulfur-containing POSS. The nanocomposites were blended with benzoxazine containing allyl group (BOZ)/epoxy composites and cured. The effect of POSS on the thermal stability and flame retardancy of the material was studied.

## 2. Experimental Section

### 2.1. Materials and Methods 

Allylamine was obtained from the Chengdu Huaxia Chemical Reagent Co. Ltd. (Chengdu, China), >95%. Paraformaldehyde was obtained from the Tianjin Fuchen Chemical Reagent Factory (AR, Tianjin, China). Phenol was obtained from the Beijing Chemical Plant (AR, Beijing, China). The SPOSS (TH-1550) was obtained from Hybrid Plastics Inc. (Hybrid Plastics®, Hattiesburg, MS, USA), >99%. DMPA was purchased from Aladdin Biochemical Technology Co. Ltd. (Shanghai, China), >99%. Tetrahydrofuran was obtained from Beijing Tongguang Fine Chemical Company (AR, Beijing, China). Epoxy resin was purchased from E-44, Beijing Tongguang Fine Chemical Company (Beijing, China).

### 2.2. Synthesis of 3-Allyl-3,4-Dihydro-2H-1,3-Benzoxazine Monomers (BOZ)

The synthesis route of BOZ is shown in Scheme 1 [1]. Paraformaldehyde (25.6 g, 0.8 mol) and allylamine (22.8 g, 0.4 mol) were stirred in toluene (60 mL) at room temperature for 30 min under a N_2_ atmosphere. Then, phenol (37.6 g, 0.4 mol) was added into the reaction system. The mixture was stirred at 80 °C for 2 h. Subsequently, the solution was cooled to room temperature, and washed using 1 mol/L NaOH solution and distilled water several times. Then, the organic layer was dried over anhydrous Na_2_SO_4_. The solution was dried under vacuum to afford a light-yellow liquid. Yield: 56 g, 80%; ^1^H NMR (600 MHz, CDCl_3_, 298 K): δ 6.80–7.13 (s, 4H, Ar–H), 5.92 (tt, 1H, C–CH=C), 5.27–5.18 (m, 2H, =CH_2_), 4.88 (s, 2H, O–CH_2_–N), 4.00 (s, 2H, Ar–O–N), 3.40 (d, 2H, N–CH_2_–C). FTIR (cm^−1)^: 3077 cm^−1^ (stretching of =C–H), 1843–1560 cm^−1^ (skeleton vibration of benzene ring), 1646 cm^−1^ (stretching of C=C), 914 cm^−1^ (symmetric stretching of C-O-C), 746 cm^−1^ (out-of-plane bending vibrations of olefinic =C–H). HRMS (MALDI-TOF): m/z 175 [M+, calcd 175].

### 2.3. Preparation of the Nanocomposites Benzoxazine POSS (SPOSS-BOZ)

Benzoxazine POSS was synthesized using the procedure presented in Scheme 2. Both the benzoxazine containing allyl group (BOZ) and SPOSS were dissolved in tetrahydrofuran (THF) and this solution was stirred at room temperature for 10 min, then the initiator of DMPA (1% eq) was added and irradiated for 5 min under UV light. Then, the resultant product was collected once dried in vacuum to obtain SPOSS-BOZ as a light-yellow powder. 

### 2.4. Preparation of the Composites SPOSS-PBZ-E

Composites (SPOSS-PBZ-E) containing polybenzoxazine, POSS, and epoxy resin were prepared by the procedure in Scheme 3 and the schematic diagram of the preparation method is shown in Figure 1. The desired amounts of SPOSS-BOZ, BOZ, and E-44 were mixed and stirred at 80 °C for 30 min. The residual solvent was removed in vacuum at 60 °C to afford homogenous mixtures. The mixture was poured into Teflon molds and cured in an air circulating oven (Yiheng Scientific Instrument Co., Ltd., Shanghai, China) by the following steps: 80 °C (1 h), 110 °C (1 h), 130 °C (1 h), 160 °C (1 h), 180 °C (1 h), and 210 °C (1 h). For these composites, the mass ratio of BOZ to E-44 was 1:1 and then we added a certain amount of SPOSS-BOZ. The cured product was transparent and had a red-wine color and the procedure was repeated to prepare other hybrid materials by varying the amount of SPOSS-BOZ (0, 5, 10, 15, and 20 wt %).

### 2.5. Measurements and Characterization 

UV-curing: A HWUV0133X three-dimensional UV curing box (Zhonghe Machinery Equipment Manufacturing Co., Ltd, Baoding, China) was used for the experiment. The wavelength of the ultraviolet light was 300–400 nm, the lamp power was 400W, and the radiation intensity was 100 mW/cm^2^.

Nuclear magnetic resonance (NMR) spectroscopy: ^1^H-NMR spectra were recorded on a Bruker Avance 600 NMR spectrometer (BRUKER OPTICS, Beijing, China) operated in the Fourier transform mode. CDCl_3_ was used as the solvent, and the solution was measured with tetramethyl silane (TMS) as an internal reference.

Fourier-transform infrared (FTIR) spectroscopy: FTIR spectra were recorded on a NICOLET 6700 IR spectrometer (BRUKER OPTICS, Beijing, China). The spectra were collected at 32 scans with a spectral resolution of 4 cm^−1^.

Mass spectrometry: Matrix-assisted laser desorption/ionization time-of-flight mass spectrometry (MALDI-TOF MS) was performed using a Bruker BIFLEX III device (BRUKER OPTICS, Beijing, China) equipped with a pulsed nitrogen laser (k ¼ 337 nm, pulse width ¼ 3 ns, and average power ¼ 5 mW at 20 Hz). Samples were measured in positive ion modes with usually 50 spectra accumulated. For the MALDI mass spectrum, we used a-Cyano-4-hydroxycinnamic acid matrix and the salts were a mixture of NaCl and KCl. 

X-ray diffraction (XRD). The samples were scanned at a speed of 8 °C/min at ambient temperature using an X-ray diffractometer (DX-2600, Rigaku corporation, Tokyo, Japan) at a generator voltage of 35 kV and a current of 25 mA. The data were collected from 2° to 40° intervals.

Thermogravimetric analysis (TGA) was performed with a NETZSCH 209 F1 thermal analyzer (NETZSCH, Bavarian, Germany) at a heating rate of 10 °C/min under nitrogen and air atmosphere, respectively. The temperature ranged from 40 to 800 °C.

Differential scanning calorimetry (DSC) curves of the EP composites were measured using a Netzsch 204 F1 differential scanning calorimeter (NETZSCH, Bavarian, Germany) with a pressure cell. Samples (5–10 mg) were tested at a heating rate of 10 °C/min and results from the second heating were in the range of 35–300 °C. 

The dynamic mechanical(DMA) properties of the cured blends were performed using a METTLER TOLEDO SDTA861 instrument (Greifensee, Switzerland) with a sample dimension of 5 × 5 × 3 mm in a controlled strain tension mode and a temperature ramp rate of 4 °C/min from −70 to 200 °C at a frequency of 1.0 Hz.

The limiting oxygen index (LOI) was obtained using the standard ASTM D 2863 procedure, which involves measuring the minimum oxygen concentration required to support candle-like combustion of plastics. An oxygen index instrument (Rheometric Scientific Ltd., (Phoenix Instruments Co., Ltd, Suzhou, China) was used on barrel-shaped samples with the dimensions of 100 × 6.5 × 3 mm

Vertical burning tests were performed using the UL 94 standard on samples with the dimensions of 125 × 13 × 3.2 mm with the CZF-5A horizontal vertical burning tester by Jiangning Analytical Instrument Factory (Phoenix Instruments Co., Ltd, Suzhou, China). In this test, the burning grade of a material was classified as V-0, V-1, V-2, or NR (no rating), depending on its behavior (dripping and burning time).

Scanning electron microscopy (SEM) and energy dispersive X-ray spectroscopy (EDX) images were taken with a Hitachi SU8020 (Beijing, China) with a 20 kV accelerating voltage. The samples were coated with thin layers of gold to make the surface conductive. 

Cone calorimeter measurements were performed at an incident radiant flux of 50 kW/m^2^, according to the ISO 5660 protocol, using a Fire Testing Technology apparatus (Phoenix Instruments Co., Ltd, Suzhou, China) with a truncated cone-shaped radiator. The specimen (100 × 100 × 3 mm) was measured horizontally without any grids. Typical results from the cone calorimeter were reproducible within ±10%, and the reported parameters were the average of three measurements.

## 3. Results and Discussion

### 3.1. Characteristics of Benzoxazine Monomers (BOZ)

Different synthesis methods can be used to prepare benzoxazine monomers. Agag and Takeichi prepared monomers with a solventless method [1]; However, the solvent method minimizes the oligomer production. The chemical structure of BOZ was confirmed by ^1^H NMR (Figure 2). The resonances of –CH*=C and =${CH}_{2}^{*}$ protons appeared at 5.9 ppm and 5.2–5.25 ppm, respectively. The resonances at 4.0 ppm and 4.87 ppm were assigned to the proton of ${Ar–CH}_{2}^{*}$–N and ${N–CH}_{2}^{*}$–O, respectively, confirming the formation of the oxazine ring and indicating the high purity of BOZ. The FTIR spectrum of BOZ provided the information shown in Figure 3a. The bands at 3077 cm^−1^ and 1646 cm^−1^ represented the characteristic peaks of =C–H* and C=C, respectively. The characteristic absorptions of the benzoxazine ring were at 916 cm^−1^ (symmetric stretching of C–O–C). Moreover, the molecular quality of P-ala was obtained in Figure 3b. Based on these results, BOZ was effectively prepared.

### 3.2. Preparation of the Nanocomposites Benzoxazine POSS (SPOSS-BOZ)

The monofunctional POSS derivative is the most useful compound for copolymerization with other monomers through diverse functional groups, and a mono-functionalized BOZ ring containing POSS (BOZ-POSS) has been prepared using different methods [32,33]. However, we synthesized SPOSS-BOZ using the thiol-ene (click) reaction, which is an effective method with almost no side reactions [34]. SPOSS and P-ala were dissolved in tetrahydrofuran (THF) and this solution was irradiated only for 5 min under UV light to prepare the mono-functionalized BOZ ring containing POSS. It can be seen from the FTIR spectrum in Figure 4 that the S–H peak at 2700 cm^−1^ disappeared after SPOSS was modified by benzoxazine. The chemical structure of SPOSS-BOZ was confirmed by ^1^H NMR (Figure 5). The resonance peaks at 6.8–7.2 ppm were assigned to the aromatic protons. The characteristic protons of the oxazine ring appeared at 4.8 ppm and 3.96 ppm (Peak c) and were assigned to –O–CH_2_–N– and –Ar– CH_2_–N–, respectively. The resonance peaks at 0.68 ppm (Peak b), 0.93 ppm (Peak a), and 1.88 ppm were caused by the seven isobutyl hydrocarbon substituents of the POSS. Next, we magnified the spectra of the allyl moiety (Figure 5B). Furthermore, the resonances of –CH*=C and =${CH}_{2}^{*}$ protons that appeared at 5.9 ppm and 5.2–5.25 ppm almost disappeared, respectively. Therefore, POSS was successfully incorporated into benzoxazine.

### 3.3. Curing Behavior of Polybenzoxazine/POSS/Epoxy (SPOSS-PBZ-E)

The thermally activated polymerization reaction of the copolymers was also studied by non-isothermal DSC, as shown in Appendix A. The oxazine ring opening highly overlapped the allyl addition polymerization exotherm at the temperature range of 170–250 °C. To eliminate this effect, the extrapolation method was used to determine the peak temperature when the heating rate β was 0, and the curing temperature range was determined. The DSC data is shown in Table 1. It can be seen from Figure 6 that when β was 0, *T*_i_, *T*_p_, and *T*_f_ were 144 °C, 210 °C, and 240 °C, respectively, and the resin system started to react in the range of 144 to 210 °C. Combined with the relevant data [35], the curing process of the resin system was determined to be 80/1 h + 110/1 h + 130/1 h + 160/1 h + 180/1 h + 210/1 h. Furthermore, the step-wise curing process of the composites was studied by FTIR spectroscopy as shown in Figure 7. As the curing temperature increased, the bands at 1649 cm^−1^ and 916 cm^−1^ decreased due to the allyl and the characteristic peaks of the oxazine ring respectively, which indicated that the ring-opening reaction and polymerization of the oxazine ring had taken place. Meanwhile, the peak at 3203 cm^−1^ was due to the phenolic hydroxyl group that occurred at 160 °C/1 h. With curing proceeding, the peak at 1224 cm^−1^ vanished because the epoxy group decreased, whereas the peak at 1012 cm^−1^ increased due to the ether bond. The reaction between the phenolic hydroxyl group and epoxy group was found. Therefore, these results proved that the copolymer had been completely polymerized [9].

### 3.4. X-Ray Diffraction Analysis

Figure 8 displays the X-ray diffraction patterns for the SPOSS and SPOSS-PBZ-E nanocomposites. The sharp diffraction peaks seen for the SPOSS indicated a highly crystal structure consistent with the literature [36,37,38,39,40]. As can be seen in Figure 8, the XRD pattern for BOZ:E itself showed one broad peak at 2θ ≈ 20°, which confirmed its amorphous structure. After the incorporation of SPOSS nanoparticles into the BOZ:E matrix, the peaks observed in the pure POSS nanoparticle completely disappeared in the case of the 5 wt % SPOSS-PBZ-E, suggesting the molecular dispersion of the SPOSS nanoparticles in the SPOSS-PBZ-E nanocomposite, and also displayed a broad peak at 2θ ≈ 20°. This indicated that SPOSS was chemically incorporated into the hybrid nanocomposite and formed a cross-linked network between the SPOSS and BOZ:E blend. However, a crystalline peak at 2θ = 7.97°, corresponding to the strong diffraction of the SPOSS monomer, was observed when the content of SPOSS was more than 10 wt %. Hence, this indicates that separate POSS domains were present in the SPOSS-PBZ-E nanocomposites.

### 3.5. Scanning Electron Microscopy Micrographs

Figure 9 displays the cross-sectional images of materials with various SPOSS-BOZ contents. The hybrid materials had a good distribution, and microphase separation was not observed in the matrix. As seen from the image of the benzoxazine/epoxy resin, the majority of the POSS particles are in the size range of 80–200 nm. However, when the amounts of SPOSS-BOZ reached 10 wt % and 20 wt %, the fracture surface was very rough. Furthermore, as shown in Appendix A and according to the results in Table 2, an EDX analysis showed the presence of POSS particles. The different contents of silicon atoms indicate the different contents of POSS particles in the whole system.

### 3.6. Thermal Properties of the SPOSS-PBZ-E Nanocomposites

TGA in the nitrogen and air atmospheres, derivative thermograms, and tan δ curves of the nanocomposites are shown in Figure 10, Figure 11 and Appendix A, respectively. A summary of the thermal properties for each nanocomposite Is listed in Table 3.

From the DMA experimental data, the glass transition temperature (*T*_g_) of the nanocomposites was slightly lower when compared with the BOZ:E matrix. As the POSS cage structure consisting of Si atoms and O atoms was bulky, this hindered the formation of the benzoxazine/epoxy resin cross-linked network to a certain extent and reduced the crosslink density [41]. The char yield of the SPOSS-PBZ-E nanocomposites increased gradually as the content of SPOSS-BOZ increased due to the increase in the silica and SiO_2_ yield that was produced by the thermal degradation of POSS. The char yield in the air atmosphere was far less than in N_2_. As can be seen from the thermograms, not only in the N_2_ but also in the air atmosphere was the weight loss curve of the BOZ:E matrix not significantly altered by the presence of SPOSS. Similar weight loss traces were found from the TGA profiles. The TGA clearly showed that the 5% mass loss temperature (*T*_5%_) of SPOSS-PBZ-E nanocomposites was slightly lower than the BOZ:E matrix except for the modified composites with a 20 wt % addition of SPOSS-BOZ in the N_2_ atmosphere Therefore, the decrease in *T*_5%_ and *T*_g_ was consistent. At a later period of the degradation process, the degradation rate slowed down with the increased POSS content and the *T*_peak_ of all the SPOSS-PBZ-E hybrid nanocomposites almost remained unchanged with respect to that of the BOZ:E matrix. The steric hindrance and interaction with the polymer chains caused by the incorporation of the POSS cages may result in a limit to the moving of polymer chains during the thermal degradation and lead to a small increase in thermal stability. Moreover, the thermal stability of the inorganic component (POSS) was very good and, during combustion, the POSS units could form a ceramic superficial layer because of the low surface energy of the siloxane structure of POSS [36]. Therefore, the char yield of the nanocomposites increased with the addition of the SPOSS-BOZ organic–inorganic hybrid. When the amount of SPOSS added was 20%, the mass residual ratio of the system increased by 31.8% and 27% compared with that before modification, which was 29.45% in N_2_ and 2.8% in air, respectively. 

### 3.7. Flame Retardancies of SPOSS-PBZ-E

The flame retardancies of the SPOSS-BOZ-E composites are presented in Table 4. Following the incorporation of SPOSS-BOZ, the LOI value of the blends increased slightly from 28.2% to approximately 28.9%. Thus, the silsesquioxane apparently played only a weak role in affecting the LOI value of the epoxy resin. However, the UL 94 vertical burn test includes the vertical burning level, the average afterflame time t_1_ and t_2_ after the first and second ignition, total afterflame time (t_1_ + t_2_), and cotton ignited by flaming particles or drops. The composites of PBZ-E without SPOSS-BOZ and 5% SPOSS-BOZ burned readily and had no rating; however, even with an incorporation of 10% of SPOSS, the burning behavior of blends altered and there was no dripping. The UL 94 V-1 rating was obtained and the total afterflame time (t_1_ + t_2_) was reduced in comparison to the 5% POSS-BOZ-E. During combustion, the SPOSS-BOZ nanocomposites formed a dense SiO_2_ layer that could prevent the penetration of combustion. When the POSS content cannot reach this level, an effective SiO_2_ layer will not be formed during combustion. 

### 3.8. Cone Calorimeter Analysis

Cone calorimetry is used as a more promising technique to compare and evaluate the fire performances of polymeric materials. The fire performances of the composites were compared using time to ignition (TTI), peak heat release rate (pHRR), average HRR (aHRR), total heat release (THR), total smoke production (TSP), and total smoke release (TSR). These parameters are reported in Table 5 for the different nanocomposites. 

TTI is used to determine the influence of flame retardant on ignitability. As shown in Table 5, the TTI of the flame-retarded epoxy resins with SPOSS-BOZ loading did not show any effective enhancement.

The HRR versus time curves for the SPOSS-BOZ-E nanocomposites are presented in Figure 12. It can be seen that all the composites burned rapidly, and the typical char-forming platform was not observed. For the neat benzoxazine/epoxy resin, the pHRR was 1156 kW/m^2^, and the pHRR values of the 5% SPOSS-PBZ-E, 10% SPOSS-PBZ-E, 15% SPOSS-PBZ-E, and 20% SPOSS-PBZ-E were 957, 962, 942, and 769 kW/m^2^, respectively. It is clear that SPOSS-BOZ apparently hinders the decomposition of the composites and improves the flame retardancy. The addition of 20 wt % SPOSS-BOZ caused reductions in the pHRR, aHRR, and THR values by about 33%, 27%, and 25%, respectively (Table 5). The reductions of these data resulted from the effect of the formation of a char barrier (Figure 13). During the cone calorimeter test, the organic group of POSS nanoparticles degrades and leaves the Si–O–Si-based ceramic layer [37,38]. As shown in Figure 13, when the content of SPOSS-BOZ increased, the residual carbon content of the benzoxazine/epoxy resin increased slightly, and when the content of SPOSS-BOZ reached 20 wt %, a thicker, continuous, and intumescent insulation layer with an integral structure was formed during combustion. However, the TSR of samples increased in the 15%SPOSS-PBZ-E and 20%SPOSS-PBZ-E samples with enlarging POSS content. 

The char residues were further analyzed with SEM-EDX analysis. The external char residues of the BOZ-E, 5% SPOSS-PBZ-E, 10% SPOSS-PBZ-E, and 15% SPOSS-PBZ-E composites and the internal and external char of the 20% SPOSS-PBZ-E were analyzed by SEM-EDX (Figure 14 and Figure 15). It can be seen that the char of BOZ-E was loose and porous, but for the nanocomposites containing SPOSS-BOZ, a dense and compact external layer can be seen in Figure 14. From Figure 15, the presence of the Si atom can be observed on the char surface with the POSS nanoparticles, and the content of Si reaches its highest value with the 20 wt % SPOSS-BOZ loaded from the EDX, but there is less silicon in the internal char. It is thought that the increase in fire protection from using SPOSS-BOZ arises from the formation of a ceramic layer due to the enrichment of silicon on the char surface. When there are enough POSS nanoparticles, it can effectively protect the combustion of internal polymers. Therefore, the flame-retardant mechanism of POSS is mainly a condensed-phase flame retardant.

## 4. Conclusions

A novel organic–inorganic hybrid containing allyl benzoxazine and polyhedral oligomeric silsesquioxane (SPOSS) was synthesized by the thiol-ene (click) reaction. The curing process of SPOSS/polybenzoxazine/epoxy has multiple polymerization mechanisms according to the oxazine ring-opening and epoxy resin (EP) polymerization, which was characterized by FTIR, and showed that the phenolic hydroxyl group was formed after the ring opening of the oxazine ring, which promotes the ring-opening polymerization of epoxy. The X-ray diffraction analysis and the SEM micrographs of the SPOSS-PBZ-E nanocomposites indicated that SPOSS was chemically incorporated into the hybrid nanocomposites in the size range of 80–200 nm. The thermal properties and the char yield of the nanocomposites increased, and the highest maximum rate of weight loss decreased with respect to the addition of the SPOSS-BOZ organic–inorganic hybrid. At the same time, the flame retardancy of nanocomposites was improved when compared with the pure benzoxazine/epoxy blend. When the amount of SPOSS reached 10 wt % or more, the UL 94 vertical burning rating reached V-1, and when the SPOSS content reached 20 wt %, both the thermal stability and the flame retardancy of the curing system were improved. In the cone calorimeter testing, the addition of SPOSS-BOZ hindered the decomposition of the composites and the addition of 20 wt % SPOSS-BOZ caused reductions in the pHRR, aHRR, and THR values of approximately 33%, 27%, and 25%, respectively. Then, the morphologies of the chars were also studied by SEM and EDX, which indicated that during the combustion process of the nanocomposites, the POSS nanoparticles accumulated on the surface and formed a ceramic layer in the exterior char residue.

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
