# Peer review of "Effects of an Organic-Inorganic Hybrid Containing Allyl Benzoxazine and POSS on Thermal Properties and Flame Retardancy of Epoxy Resin"

_polymers, 2019, doi:10.3390/polym11050770_

Round 1
Reviewer 1 Report
The paper from Liu et al. reports on the synthesis (using the thiol-one click reaction) of a O/I hybrid containing allybenzoxazine and POSS and its use as flame retardant in a benzocaine/epoxy resin.
The idea shows some novelty, but the conclusions are only partially supported by the experimental data. Therefore, major revision is suggested. In particular:
- UV-curing: plase specify in the Experimental the type of UV-lamp and the radiation intensity (in mW/cm2)
- page 5, line 156-159: please add the conditions adopted for performing TG analyses in air
- cone calorimetry tests: why did the authors perform forced combustion tests at 50kW/m2? Did they previously tried with 35 kW/m2?
- cone calorimetry tests: what about the smoke parameters? A table collecting these data should be added to the text and the data commented
- Table 4: please add the final residue
- It could be reasonable to perform EDX analyses on the cross-sections of the prepared nanocomposites (not on the residues), hence mapping Si element distribution within the epoxy matrix.
Author Response
Response to Reviewer 1 Comments
Dear Reviewer:
Thanks for your cautious review and the comments concerning our manuscript entitled “Effects of organic-inorganic hybrid containing allyl benzoxazine and POSS on thermal and flame retardancy of epoxy resin” (ID: polymers-486624). These comments are all valuable and helpful for revising and improving our paper. The revised portions are highlighted in red in the paper. The responses to the comments are as flowing:
Point 1: UV-curing: please specify in the Experimental the type of UV-lamp and the radiation intensity (in mW/cm2)
Responds 1:We have specified experimental type of UV-lamp and the radiation intensity (in mW/cm2),It can be seen from page 5,line 145 to line 147.
Point 2: page 5, line 156-159: please add the conditions adopted for performing TG analyses in air
Responds 2: We have added this condition of TGA in the page 5,line 162 to line 164.
Point 3: Cone calorimetry tests: why did the authors perform forced combustion tests at 50kW/m2? Did they previously tried with 35 kW/m2?
Responds 3: As you said, both the heat flux of 35 kW/m2 and 50 kW/m2 can be used as test conditions in the cone calorimeter. But, when I choose the test condition, I consider the flame temperature corresponding to 50 kW/m2 is about 700-800 °C, which close to the temperature of a real fire. So, the heat flux of 50 kW/m2 is the preferred test condition.
Point 4: cone calorimetry tests: what about the smoke parameters? A table collecting these data should be added to the text and the data commented.
Responds 4: We have added the data of total smoke production (TSP), total smoke release (TSR) and final residue in the text and Table 5, which can be seen form page 15, line 347 to 367.
Point 5:It could be reasonable to perform EDX analyses on the cross-sections of the prepared nanocomposites (not on the residues), hence mapping Si element distribution within the epoxy matrix.
Responds 5: The SEM-EDX of the cross-section of SPOSS-PBZ-E nanocomposites were re-tested, and Si element distribution within the epoxy matrix was obtained. The result can be seen from page 11, line 269 to line 279. As seen from image of benzoxazine/epoxy resin,majority of the POSS particles are in the size range of 80-200 nm. Furthermore, the EDX analysis show the presence of POSS particles and the different contents of silicon atom indicate that the different contents of POSS particles in the whole system.
Point 6: Extensive editing of English language and style required
Responds 6: For English questions, we checked our manuscript through English editing service provided by MDPI. So, there should be no problems with English grammar.
Once again, thank you very much for your comments and suggestions.
Best regards

Reviewer 2 Report
First of all, the authors should provide a clear version of the manuscript with modification highlighted in red. The manuscript is now under tracking mode, which is very hard to read. The response letter should also list the review comments and corresponding response as well as the changes in the manuscripts. The authors did the experiments as suggested by the reviewer. However, these figures have a lot of issues. For example, figure 3b, the numbers and unit on x-axis are missing. On Figure 5, what is the difference between (a) and (b)? The captions for (a) and (b) are missing.
Considering those problems, I would recommend the authors to provide a clear version for reviewing again.
Author Response
Response to Reviewer 2 Comments
Dear Reviewer:
Thanks for your patient and cautious comments concerning our manuscript entitled “Effects of organic-inorganic hybrid containing allyl benzoxazine and POSS on thermal and flame retardancy of epoxy resin” (ID: polymers-486624). These comments are all valuable and helpful for revising and improving our paper. I am very sorry that the last version of the manuscript is not clear. The revised portions are highlighted in red in the paper. I have submitted a new revised report, which also included all the comments:
Point 1: TG analyses should also be performed in air, and the obtained results commented:
Responds 1:TG analyses have been performed in air, and the results comparing in N2 were discussed from page 12, line 290 to line 320 in the manuscript.
Point 2:XRD analyses and SEM/TEM microscopy could better show the morphology of the obtained hybrids
Responds 2:We added the X-ray diffraction and SEM-EDX analysis for SPOSS-PBZ-E nanocomposites and discussed the dispersion of SPOSS-BOZ in matrix. It can be seen from page 10, line 252 to line 289. As seen from image of benzoxazine/epoxy resin,majority of the POSS particles are in the size range of 80-200 nm. Furthermore, the EDX analysis show the presence of POSS particles and the different contents of silicon atom indicate that the different contents of POSS particles in the whole system.
Point 3:Finally, and this is a very weak point, the characterization of the fire retardancy lacks of foirced combustion tests
Responds 3:According to your suggestions, we also added the cone calorimetry tests and analyze the carbon layer structure by SEM-EDX. It can be seen from page 14, line 341 to page 15, line 389 in the manuscript.
Point 4:Figure 3b, the numbers and unit on x-axis are missing. On Figure 5, what is the difference between (a) and (b)? The captions for (a) and (b) are missing.
Responds 4:The new revised manuscript have been modified and checked carefully.
Point 5:English language and style are fine/minor spell check required
Responds 5: For English questions, we checked our manuscript through English editing service provided by MDPI. So, there should be no problems with English grammar.
Once again, thank you very much for your comments and suggestions.
Best regards

Round 2
Reviewer 1 Report
The authors have revised the manuscript according to the Reviewers ' comments and suggestions. Now the manuscript seems suitable for publication.
Reviewer 2 Report
The authors made the changes suggested by the reviewer. I would recommend it to be published in polymers
This manuscript is a resubmission of an earlier submission. The following is a list of the peer review reports and author responses from that submission.
Round 1
Reviewer 1 Report
The authors reported the synthesis and flame retardancy property of polybenzoxazine/ POSS nanocomposites. However, the polybenzoxazine/ POSS nanocomposites has been reported previously, using a similar strategy (Polymer 2004, 45 (18), 6321-6331.). The synthesis that using thiol-ene chemistry didn’t bring new insight into the system. Moreover, the authors didn’t discuss the advantage of their synthesis method compared with other papers. The authors also didn't provide strong evidence and characterization for the successful thiol-ene reaction, such as NMR. Finally, the writing needs to be significantly improved. Therefore, I can’t recommend this paper to be published in polymers.
1. English should be carefully checked throughout the manuscript. For example line 52, “The incorporation of polyhedral oligomeric silsesquioxane (POSS) into BZ monomers can improve the thermal or mechanical properties of polybenzoxazine resins have been studied.” line 58, “a Organic-inorganic hybrid benzoxazine was prepared by thiol-ene click chemistry reaction between the and sulfur-containing POSS.”
2. The term “EZ” and “BOZ” should be defined.
3. The microstructure of the nanocomposites is important and related to their mechanical properties. So need to be characterized too.
4. A lot of related works and reviews about the polybenzoxazine/ POSS nanocomposites was not included in the paper (for example, Polymers 2016, 8 (6), 225). Additionally, the thiol-ene chemistry in POSS has been well studied and deserve the discussion in the introduction. (J. Mater. Chem., 2011,21, 12753-12760; ACS Macro Lett., 2012, 1 (7), pp 834–839; Polymer 2017, 125, 303-329; Polymer 2018, 145, 324-333; Chemical Engineering Journal 2018, 332, 150-159.)
5. The fire performances should be evaluated by a series of techniques, such as heat release rate and mass loss. For example, SEM-EDX technique allows to obtain the char residues, see ref. Polymer Degradation and Stability 2018, 149, 96-111.
Author Response
Dear Reviewer:
Thanks for your cautious review and the comments concerning our manuscript entitled “Effects of organic-inorganic hybrid containing allyl benzoxazine and POSS on thermal and flame retardancy of epoxy resin” (ID: polymers-446140). These comments are all valuable and helpful for revising and improving our paper. The revised portions are highlighted in yellow in the paper. The responses to the comments are as flowing:
(1)According to the article of Polymer 2004,45(18),6321-6331,Lee Y J et al. synthesized mono-functionalized BOZ ring containing POSS (POSS-BOZ) using two approaches: (i) the preparation of vinyl-terminated BOZ and then hydrosilylation with POSS and (ii) condensation of a primary amine-containing POSS (amine-POSS) with formaldehyde and phenol in THF solution at 90 °C to obtain POSS-BOZ. theses synthesis steps are complicated and rigorous, especially in the case of platinum as a catalyst, requiring a water-free environment. Comparing with their work, the thiol-ene click reaction to prepare the POSS-BOZ nanocomposites in our manuscript is more sample (simple?). We are very sorry for our negligence of NRM Characterization of Conversion Rate of SOSS-BOZ,and we have re-characterized it using 1H NMR in the revised edition, the conversion ratio of SPOSS-BOZ is about 72.5%. It’s more effective than above methods. Also, we have re-discussed the concerning of yours in introduction. As to the English writing, we did also modify the text to meet requirements, if still lack of expression, we would ask for help from professional grammar service.
(2) We have carefully checked the English question of the article and corrected some mistakes.
(3) We are very sorry for our incorrect writing of “BZ” and “BOZ”, they are the same meaning. And we have corrected them. In the revised edition, “BOZ” has been defined as “benzoxazine”
(4) we are currently very sorry that we are unable to provide the data of microstructure, we were requested to modify the manuscript draft within only 10 days, but this duration happened to collide with the Chinese Lunar New Year, during which our school is in the holiday stage. The time for us is too short and rigorous to obtain all the required data. We feel deeply sorry about we cannot provide the required data immediately in this revised edition.
However, according to the page 2, line 78:” Sulfur-based POSS is introduced into olefin material [29] or epoxy resin [30] by thiol-ene and thiol-epoxy click reaction, respectively, and they all have an excellent dispersibility.” Both article (Journal of Materials Chemistry, 2011, 21(34): 12753-12760) [29] and (Polymer, 2018, 145: 324-333) [30] used sulfur-based POSS to prepare nanocomposites, and both showed good dispersion. So, sulfur-based POSS has good dispersibility in epoxy and benzoxazine, our method is like them. Thus, these can be used as indirect proofs of good dispersion.
(5) Considering your kind suggestions, we have revised the introduction referring to all the related works and reviews about the polybenzoxazine/ POSS nanocomposites recommended by reviewer. The revised portions are highlighted in yellow in the paper.
(6) For the last comment, this study was designed to synthesize SPOSS/BOZ nanocomposites and apply them to benzoxazine/epoxy composites to investigate their macroscopic effects on flame retardancy and thermal properties. We focus more attention on the macroscopic effects rather than the detailed forced combustion test, so we didn't perform the cone calorimetry tests in this manuscript.
However, we did plan to discuss the flame retardancy mechanism in the further studies, in which the TGA-FTIR, XPS and SEM analysis will be used to characterize the gaseous products and the condensed residue in thermal decomposition, and the micro-structure of the chars from cone calorimeter tests. If you insist on the data requirement of cone calorimeter tests, we would be glad to update our manuscript in the next edition (so please persuade the editor to give us some more time).
Once again, thank you very much for your comments and suggestions.
Best regards
Reviewer 2 Report
This manuscript presents the synthesis of POSS-containing polymer composites with an interesting method of POSS incorporation. The author used thiol-ene click chemistry to make benzoaxine monomer and copolymerized them with epoxy at various POSS contents. The manuscript is well-written. It just have some questions 1) I know that the authors showed the FTIR of SPOSS-BPZ, but the thiol peak is already small to start with. Given that allyl-thiolene reaction is somewhat less reactive than other ene-bond, I wonder what is the conversion level of SPOSS-BOZ by NMR? 2) In page 7, line 194, it should be tan delta. Delta with a greek letter.
Author Response
Dear Reviewer:
Thanks for your cautious review and the comments concerning our manuscript entitled “Effects of organic-inorganic hybrid containing allyl benzoxazine and POSS on thermal and flame retardancy of epoxy resin” (ID: polymers-446140). These comments are all valuable and helpful for revising and improving our paper. The revised portions are highlighted in yellow in the paper. The responses to the comments are as flowing:
(i)We are very sorry for our negligence of NRM Characterization of the conversion ratio of SPOSS-BOZ,and we have re-characterized it by using H NMR spectrum. the conversion ratio of SPOSS-BOZ is about 72.5%. (page 7, line 200)
(ii)We are very sorry for our incorrect writing of tan delta, and we have corrected it with tan δ. (page 9, line 235 and page 10, line 254)
We tried our best to improve the manuscript and made some revisions in the manuscript. All the revisions have been highlighted in yellow.
Once again, thank you very much for your comments and suggestions.
Best regards
Reviewer 3 Report
The paper from Liu et al. reports on the synthesis and characterization of O/I hybrids containing allyl benzoxazine and POSS; the hybrids are synthesized through a thiol-ene (click chemistry) reaction.
The paper shows some novelty and could be of interest for the readers. However, it need a deep revision in order to become suitable for publication in Polymers. In particular:
- line 23: please replace "thermal" with "thermo"
- Introduction: it should be better point out the step-forward of the proposed research with respect to the already existing scientific literature
- TG analyses should also be performed in air, and the obtained results commented
- XRD analyses and SEM/TEM microscopy could better show the morphology of the obtained hybrids
- Finally, and this is a very weak point, the characterization of the fire retardancy lacks of foirced combustion tests (i.e. cone calorimetry tests): in fact, the authors focused on flammability tests (i.e. (LOI and UL94V) only
Author Response
Dear Reviewer:
Thanks for your cautious review and the comments concerning our manuscript entitled “Effects of organic-inorganic hybrid containing allyl benzoxazine and POSS on thermal and flame retardancy of epoxy resin” (ID: polymers-446140). These comments are all valuable and helpful for revising and improving our paper. The revised portions are highlighted in yellow in the paper. The responses to the comments are as flowing:
(i)We are very sorry for our incorrect writing of “thermal”, and we have replaced it with “thermo”. (page 1, line 24 and page 5, line 145)
(ii) We have re-written the second half of introduction according to your suggestion. We have re-discussed the preparation of POSS/BOZ nanocomposites and the application of the click reaction in the preparation of POSS/BOZ composites, and we have compared the differences between them and the thiol-ene clicks we used. Obviously, the thiol-ene click reaction is much simple and efficient. (page 2, line 50-82)
(iii) Considering your third suggestion, we are very glad to supply the TGA data, XRD analyses and SEM/TEM microscopy as your suggestion, which recognized us these data are very valuable and helpful for revising and improving our study, as well as the important guiding significance to these researches. However, we were requested to modify the manuscript draft within only 10 days, but this duration happened to collide with the Chinese Lunar New Year, during which our school is in the holiday stage. The time for us is too short and rigor to obtain all the required data. We feel pleased that you accept out research bur also deeply sorry about we cannot provide the required data immediately in this revised edition. However, we hope you to suggest the editor to give us another minor revision request to us, as thus we would upload the required data in the latest edition.
(iv) For the last comment,this study was designed to synthesize SPOSS/BOZ nanocomposites and apply them to benzoxazine/epoxy composites to investigate their macroscopic effects on flame retardancy and thermal properties. We focus more attention on the macroscopic effects rather than the detailed forced combustion test, so we didn't perform the cone calorimetry tests in this manuscript.
However, we did plan to discuss the flame retardancy mechanism in the further studies, in which the TGA-FTIR, XPS and SEM analysis will be used to characterize the gaseous products and the condensed residue in thermal decomposition, and the micro-structure of the chars from cone calorimeter tests. If you insist on the data requirement of cone calorimeter tests, we would be glad to update our manuscript in the next edition (so please persuade the editor to give us some more time).
Once again, thank you very much for your comments and suggestions.
Best regards
Round 2
Reviewer 1 Report
The authors claimed that they don't have enough time to conduct the characterization suggested by the reviewers. Therefore, those important data are still missing. So I can't recommend it to be published in its current statues. But I am happy to review it again after the further revision.
Reviewer 3 Report
The revised version still lacks of fundamentals data, the authors could not add to the revised text.
Furthermore, forced combustion tests (i.e. cone calorimetry tests) are really needed, as flammability tests are not enough to describe the flame retardant behavior of the synthesized systems. Therefore, major revision is still needed.